# Comparison of Selected Dimensionality Reduction Methods for Detection of Root-Knot Nematode Infestations in Potato Tubers Using Hyperspectral Imaging

**DOI:** 10.3390/s22010367

**Published:** 2022-01-04

**Authors:** Janez Lapajne, Matej Knapič, Uroš Žibrat

**Affiliations:** Plant Protection Department, Agricultural Institute of Slovenia, 1000 Ljubljana, Slovenia; janez.lapajne@kis.si (J.L.); matej.knapic@kis.si (M.K.)

**Keywords:** hyperspectral imaging, dimensionality reduction, LDA, PLS, PCA, RandomForest, ReliefF, XGB, *Meloidogyne*, *Solanum tuberosum*

## Abstract

Hyperspectral imaging is a popular tool used for non-invasive plant disease detection. Data acquired with it usually consist of many correlated features; hence most of the acquired information is redundant. Dimensionality reduction methods are used to transform the data sets from high-dimensional, to low-dimensional (in this study to one or a few features). We have chosen six dimensionality reduction methods (partial least squares, linear discriminant analysis, principal component analysis, RandomForest, ReliefF, and Extreme gradient boosting) and tested their efficacy on a hyperspectral data set of potato tubers. The extracted or selected features were pipelined to support vector machine classifier and evaluated. Tubers were divided into two groups, healthy and infested with *Meloidogyne luci*. The results show that all dimensionality reduction methods enabled successful identification of inoculated tubers. The best and most consistent results were obtained using linear discriminant analysis, with 100% accuracy in both potato tuber inside and outside images. Classification success was generally higher in the outside data set, than in the inside. Nevertheless, accuracy was in all cases above 0.6.

## 1. Introduction

Quarantine pests are of major importance for agriculture and the food industry, and are being officially monitored and controlled [1]. Among these, root-knot nematodes (RKN) of the genus *Meloidogyne* present the most destructive group of plant-parasitic nematodes. They can infest a broad range of host plants, and are alone responsible for approximately 5% of global crop losses. These are soil-borne parasites, where they infest the host plants’ root system and cause non-specific symptoms on above-ground parts of plants. Furthermore, they can cause latent (asymptomatic) infestations in potato tubers [2], which pose an additional threat in seeding material, as they could be spread over larger areas quite quickly. The parasite *M*. *luci* has been originally described by Carneiro et al. from samples from Brazil, Chile, and Iran [3], and has since been found several times in Europe as well [4]. Even though *M*. *luci* belongs to the group of tropical RKNs, it can survive winter in fields under temperate and Mediterranean climates [5]. It is therefore considered an emerging pest in Europe and was included in the alert list of harmful organisms in 2017 [6].

Since RKNs cause non-specific symptoms, laboratory diagnoses are required for accurate identification. Traditionally RKN species are identified morphometrically, and by analysing dehydrogenase and esterase isozyme phenotypes [7]. These methods require the isolation of mature females from plant tissue, making them unsuitable for a large number of samples. First visible symptoms are presented as reduced plant growth. Infections start in small areas of the crop, but can over the years become full field infestations if not appropriately handled. Particularly latent infestations of potato seed tubers have the potential to facilitate this process and lead to infestations over larger areas in just one or two seasons. These characteristic of RKN infestations show a clear need for detection of infestations in early stages [8].

Precision agriculture helps reduce the spread of diseases, and includes well-established practices to mitigate losses [9]. The plant immune system reacts to stressors by changing their biophysical and biochemical makeup, which in turn affects their spectral properties [10]. Currently the most common remote sensing method, used for plant diseases detection, is hyperspectral imaging (HSI) [1]. Nowadays, HSI is used in various applications, such as biotechnology, agriculture, environmental monitoring, and chemistry [11]. In HSI, reflected light is captured and data stored in several spectral bands, with a high spectral resolution (bandwidths are typically around 4–5 nm). Consequently, a large number of spectral bands are acquired at each capture, for the entire spectrum the sensors record (typically between 400 and 2500 nm). 

Hyperspectral imaging has been used extensively for assessing plant root and tuber quality. The published research can generally be divided into three interlinked groups, according to the investigated properties: (1) physical properties (e.g., colour and texture), (2) chemical constituents (e.g., proteins and polysaccharides), and (3) pest and disease detection. Research into the latter is mostly focused on early detection of infections and infestations in above-ground parts of plants [12]. Biotic and abiotic stressors can cause changes in spectral signatures, by triggering various defense mechanisms, such as production of specific metabolites, induction of hypersensitive reactions, and changes in plant tissues [7]. The most extensive use of remote sensing of potato tubers has been for quality assessment, e.g., detection of defects [13] and bruises [14], and chemometric analyses, such as sugar [15], cellulose and starch content [16]. On the other hand, only a handful of studies deal with pest and disease detection in potato tubers. For example, Dacal-Nieto et al. and Huang et al. used hyperspectral imaging and support vector machines to detect hollow heart disease [17,18], and Zhou et al. used partial least squares as a pre-processing step in linear discriminant analysis to detect blackheart [19]. A partial least squares discriminant analysis approach was used by Garhwal et al. to detect zebra chip disease [20], and Al Riza et al. used a combination of genetic algorithms for feature selection and partial least squares to detect common scab [21]. But results aren’t always so clear-cut, as Zhao et al. found that infrared and thermal imaging did not distinguish between healthy and *Liberibacter solanacearum* infected tubers in storage [22].

Hyperspectral data shows a high level of collinearity between spectral bands, leading to high redundancy and decreases the signal-to-noise ratio. Furthermore, the data sets are of high dimensionality, which increases the difficulty of knowledge discovery and pattern recognition. Dimensionality reduction is therefore an obligatory and crucial step in HSI data pre-processing [10].

With dimensionality reduction (DR) methods we retain the descriptive power of the data, but reduce the number of dimensions. This process removes some patterns in the data, but the features of interest remain. The large number of spectral bands can cause reduced discriminating ability of the HSI features. This problem is especially severe when the available training set consists of a small amount of samples (referred as the curse of dimensionality). Dimensionality reduction is therefore a crucial step that transforms the data to lower dimensional space, while preserving relevant information [23]. Generally, DR methods can be grouped into two sets: (1) feature extraction, and (2) feature selection. Feature extraction methods transform the whole feature space to a lower dimensional one, while feature selection approach picks out the most significant features from the whole feature space. Furthermore, contrary to feature extraction, feature selection preserves physical characteristics of the original feature space. But, feature selection information is lost, as features are removed from further analysis. Furthermore, feature extraction is less prone to overfitting and often results in better classification accuracy [23,24]. However, there is no standard approach which would yield the best possible result for any specific dataset [25].

In recent years researchers have shown big interest in developing new methods and tools for processing of hyperspectral data. Trends show that many authors decide to choose feature selection over feature extraction DR algorithms. However conventional feature extraction algorithms are still present, due to a better performance in some cases. Moghimi et al. tested the performance of NaCl treated wheat with different feature selection methods [11]. They aggregated all with ensemble method to increase robustness and accuracy. The transformed data were classified using quadratic discriminants analysis (QDA) and validated using 5-fold cross-validation. Similarly, AlSuwaidi et al. also used feature selection method for crop disease detection [26]. As classification method they used support vector machines (SVM). Moghadam et al. showed that SVM is one of the best classification algorithms used for plant disease detection due to its generalization ability [1]. Their feature extraction method is used based on probabilistic topic modelling. Collected features with reduced dimensionality were pipelined to Latent Dirichlet Allocation model for plant leaves disease detection. On the other hand, Jin et al. directly used convolutional neural networks (CNN) for classification, without any dimensionality reduction [27]. Feature extraction problems for HSI are oftentimes solved by using convolutional neural networks [28,29]. However, CNNs usually need more training data in comparison with conventional methods [28,30], limiting their applicability.

This study was motivated by the need to find a dimensionality reduction method for detection of *M. luci*-infested potato tubers using hyperspectral imaging. The DR method ses were twofold: (1) DR methods can achieve good classification accuracy even with should enable accurate identification in combination with support vector machines (SVM), with only one extracted feature, or a very limited number thereof. Our hypothesis only one feature, and (2) data from the outside of tubers will achieve better classification success. Since only a limited amount of information is available about spectral and chemical effects of nematode infestations on potato tubers, we tested two tuber processing methods. We selected six dimensionality reduction methods, three from each group: Partial least squares (PLS), Linear discriminant analysis (LDA), Principal component analysis (PCA), RandomForest (RF), ReliefF (RFF), and Extreme gradient boosting (XGB). Of these, LDA provided the best results, as it achieved the highest classification accuracy in both external and internal images of potato tubers. All DR methods achieved better success with data from outside of tubers, except for LDA, where the results were equal.

## 2. Materials and Methods

### 2.1. Tuber Cultivation and Preparation

The tubers were obtained from an experiment on potato (*Solanum tuberosum* cv., variety Desiree) infestation with *M. luci,* which was established from June to September 2018 in a glasshouse at the Agricultural Institute of Slovenia. A total of 20 day-old plants were transplanted to 13 cm-diameter pots (V = 1 L) and supported with 1 m plastic-coated stakes. The substrate of 10 randomly selected plants was inoculated with *M. luci* at the beginning of the experiment [31]. Roots of tomato plants, infested by *M. luci* (i.e., egg-masses were visible on the root surface; the parasites were from the collection at the Agricultural institute of Slovenia) were cut into pieces and mixed. A subset of infested roots was then weighed and nematode eggs were collected in suspension, in accordance with Hussey & Barker [32]. The number of eggs was determined visually under a stereomicroscope (Nikon SMZ800). Infested roots were introduced to the substrate to a final concentration of 250 × 10^3^ eggs/plant. The presence of *M. luci* was confirmed with isoenzyme analysis [4]. The microplot experiment was completed at the end of the growing season at 97 days after inoculation.

Potato tubers were harvested at the end of the growing season in 2018 and stored in boxes in a dark storage room with ventilation and a temperature of 18 ± 2 °C for the incubation period, until further processing. Tubers from infested pots were visually checked for signs of infection (surface galls) (Figure 1). Visibly decaying tubers were excluded from further analysis. The tubers were divided into two groups, inoculated and healthy, of 5 tubers each.

### 2.2. Hyperspectral Image Acquisition

For hyperspectral imaging tubers were sliced in half and placed on a black background. This way both the outside and inside of tubers could be imaged simultaneously. Reflectance spectra in 448 bands in the VNIR (visible to near infrared) and SWIR (short-wave infrared) regions were acquired using two Norsk Elektro Optikk AS (Oslo, Norway) pushbroom hyspex cameras, VNIR-1600 (400–988 nm, 160 bands, bandwidth 3.6 nm, spatial resolution at 1 m distance 0.1 mm) and SWIR-384 (950–2500 nm, 288 bands, bandwidth 5.4 nm, spatial resolution 0.25 mm), mounted vertically above the samples at a distance of 1 m. The samples were illuminated with two calibrated halogen lamps with homogeneous light intensity between 400 and 2500 nm, placed above the samples next to the cameras. The lamps were switched on 15 min before image acquisition to stabilize the light source’s temperature drift and establish spatial uniformity of illumination [33]. A calibrated diffuse grey reference plate with 20% reflectance (SphereOptics, Herrsching, Germany) was included in each image and used to calculate reflectance. The signal-to-noise ratio was increased by scanning each line three times and calculating the average. Hyperspectral images were radiometrically calibrated to radiance units (W sr^−1^m^−2^).

### 2.3. Pre-Processing and Analysis

The data analysis workflow consisted of five stages (Figure 2). First, radiometrically corrected images were loaded into working memory. Second, images were segmented and these segments were then used to calculate reflectance values and mean spectra of each sample. The segmented image of each tuber was then divided into six equal parts. Reflectance values and mean spectra for each of these sub-segments were extracted. Then, data was split into training (4 potatoes) and test sets (1 potato). This process was repeated 5 times. In the next stage, we applied dimensionality reduction algorithms to extract the most relevant, features. In addition to DR methods, we also included a data set without any dimensionality reduction (labelled “None”). In the last stage, the chosen features were tested using support vector machine classification.

#### 2.3.1. Segmentation

Image segmentation was performed using spectral information divergence (SID). It uses a divergence measure to match HSI image pixels to reference pixels [34]. In HSI each pixel consists of multiple values which form a discrete signal. For j-th pixel we can write:(1)$$x^{\left(j\right)}={\left(x_{1},x_{2},\ldots ,x_{D}\right)}^{T}$$
where $x_{i}$ represents the value of spectral band $B_{i}$ acquired at wavelength $\lambda_{i}$. Index D represents a number of spectral bands. Probabilistic measure $p_{i}$ can be calculated for each element $x_{i}$. Probabilities for all elements are then written into vector of probabilities **p**:(2)$$\begin{array}{cc} p_{i} & =p\left(x_{i}\right)=\frac{x_{i}}{\sum_{i=1}^{D}x_{i}}\\ p\left(x^{\left(j\right)}\right) & ={\left(p_{1},p_{2},\ldots ,p_{D}\right)}^{T} \end{array}$$

Relative entropy can be calculated between p and q probability vectors with Kullback–Leibler information measure:(3)$$KL\left(\;p\left(x^{\left(j\right)}\right)\;\left|\;\right|\;\;q\left(r^{\left(C_{k}\right)}\right)\right)=\overset{D}{\underset{i=1}{\sum }}p_{i}\cdot log\frac{p_{i}}{q_{i}}$$
where $q\left(r^{\left(C_{k}\right)}\right)$ represents probability measure for reference vector r of k-th segmentation class. In our case, possible segmentation classes are included in a set: $C_{k}\in \left\{\mathrm{potato}\;\mathrm{tuber},\mathrm{background},\mathrm{reference}\;\mathrm{panel}\right\}$. Reference vectors for each segmentation class separately are constructed from manually selected area of pixels. An area for segmentation class $C_{k}$ is defined as: $S_{k}=\left(x^{\left(1\right)},\;x^{\left(2\right)},\ldots ,\;x^{\left(N\right)}\right)$ where N represents number of pixel vectors included in area corresponding to segmentation class k. Reference values can be calculated from pixels for each spectral band:(4)$$r_{i}=\frac{1}{N}\overset{N}{\underset{j=1}{\sum }}x_{\mathrm{ij}}\;i=\left(1,2,\ldots ,D\right)$$

Reference vector is then defined as:(5)$$r^{\left(C_{k}\right)}={\left(r_{1},r_{2},\ldots ,r_{D}\right)}^{T}$$

Probability vector $q\left(r^{\left(C_{k}\right)}\right)=\left(q_{1},q_{2},\ldots ,q_{D}\right)$ can then be calculated by (3) for all segmentation classes. When reference vectors are known, SID values can be calculated [35]:(6)$$SID\left(\;p\left(x^{\left(j\right)}\right),\;q\left(r^{\left(C_{k}\right)}\right)\right)=KL\left(\;p\left(x^{\left(j\right)}\right)\;\left|\;\right|\;\;q\left(r^{\left(C_{k}\right)}\right)\right)+KL\left(\;q\left(r^{\left(C_{k}\right)}\right)\;\left|\;\right|\;p\left(x^{\left(j\right)}\right)\;\right)$$

Equation (6) assigns divergence value to each pixel constructing the HSI image. Pixels from an HSI image are classified to segmentation class $C_{k}$ with smallest divergence value. The greater the similarity of pixel to reference signal, the smaller the value of divergence. With the use of additional thresholding it is possible to fully separate predefined segmentation classes. Thresholding values were chosen with trial-and-error approach. Segmentation masks for each class can then be built based on calculated divergence values (Figure 3).

#### 2.3.2. Preparation of Features

Feature vectors can be created from segmentation masks. Pixels that correspond to the same segmentation class $C_{k}$ define observing object $o^{\left(z\right)}=\left(x^{\left(1\right)},\;x^{\left(2\right)},x^{\left(j\right)},\ldots ,\;x^{\left(N\right)}\right){\;}^{T}$, where N represents number of connected pixels located within the segmentation mask. Feature vector O is calculated from defined objects with arithmetic mean for D spectral bands. For z-th object it can be calculated using following equations:(7)$$\begin{array}{cc} O_{i}^{\left(z\right)} & =\frac{1}{N}\overset{N}{\underset{j=1}{\sum }}x_{\mathrm{ij}}^{\left(z\right)}\;i=\left(1,2,\ldots ,D\right)\\ O^{\left(z\right)} & ={\left(O_{1},O_{2},\ldots ,O_{D}\right)}^{T} \end{array}$$

Spectral reflectance was calculated for tuber data, using a 50% grey reference panel. For each HSI image we calculate feature vector $O^{\left(0\right)}$, which represents the reference panel. Reflectance values of reference panel were provided by the manufacturer (SphereOptics, Germany). We assigned those values to vector $R^{\left(0\right)}$. For arbitrary object z, in the same image where $O^{\left(0\right)}$ was calculated, converted feature vector $R^{\left(z\right)}$ can be calculated as:(8)$$\begin{array}{cc} R_{i}^{\left(z\right)} & =R_{i}^{\left(0\right)}/O_{i}^{\left(0\right)}\cdot O_{i}^{\left(z\right)}\;i=\left(1,2,\ldots ,D\right)\\ R^{\left(z\right)} & ={\left(R_{1},R_{2},\ldots ,R_{D}\right)}^{T} \end{array}$$

An important note to add is that object of reference panel $o^{\left(0\right)}$ is filtered before usage in Equation (8). The reason is to remove outliers for more accurate calculation of feature vector $O^{\left(0\right)}$. Outliers are filtered with median absolute deviation (MAD). Values outside ±2MAD were removed before further calculation. MAD can be calculated by Equation (9), for each spectral band: $i=\left(1,2,\ldots ,D\right)$. In equation $x^{\left(i\right)}$ represents vector of all pixels at i-th spectral band. Value labeled as ${\tilde{x}}^{\left(i\right)}$ represents median value of this vector. With I raw vector of ones is labelled.
(9)$$\begin{array}{cc} x^{\left(i\right)} & ={\left\{o_{\mathrm{ij}}^{\left(0\right)}|i\in D,1\leq j\leq N\right\}}_{1\times N}\\ {\tilde{x}}^{\left(i\right)} & =\mathrm{median}\left(x^{\left(i\right)}\right)\\ {\mathrm{MAD}}^{\left(i\right)} & =\mathrm{median}\left(\left|x^{\left(i\right)}-{\tilde{x}}^{\left(i\right)}I\right|\right) \end{array}$$

The feature vector $R^{\left(z\right)}$ was then smoothed using a Savitzky–Golay filter to emphasize small spectral variations the same way as in Schafer [36]. Savitzky–Golay filter is based on local least squares polynomial approximation. It was shown that it reduces noise while maintaining shape and important information in a feature vector [26]. In this study the filter window length was 15, polynomial order was 2 and second order derivatives were used. Parameters were chosen with regard to the highest exhibition of performance evaluated with classification accuracy.

##### Dimensionality Reduction

In this paper three feature extraction methods (Principal component analysis, Linear discriminant analysis, and Partial least squares), and three feature selection methods (RandomForest, Extreme Gradient Boosting, and ReliefF) are taken into consideration. Main reason for pre-processing the data with the use of stated algorithms is to reduce the number of dimensions in initial space. From all algorithms we extracted only the most prominent features.

##### Principal Component Analysis

Principal component analysis is an unsupervised linear transformation technique used in machine learning applications and multivariate statistics. It is widely used across different fields, most prominently for feature extraction, dimensionality reduction and visualization. It helps identify patterns in data based on the correlation between features. PCA aims to find the directions of maximum variance in high-dimensional data and projects it onto a new subspace with equal or fewer dimensions than the original one [37]. This is achieved based on a covariance matrix formulation of centered and normalized data. Axes of original coordinate system are transformed so that newly created axes describe maximal covariance of the data. Each axis is described by an eigenvector, whose variance corresponds to its eigenvalue. Eigenvectors and eigenvalues of covariance matrix Σ can be calculated using Singular value decomposition (SVD), which can be written as:(10)$$\Sigma =\frac{1}{n-1}X^{T}X=P\Lambda P^{T}=\overset{m}{\underset{j=1}{\sum }}\lambda_{j}p_{j}p_{j}^{T}{}^{\;}$$

In Equation (10) data instances are included in matrix X∈ℝn×m, where n represents number of data instances and m number of attributes in each instance. Matrix P ∈ℝm×m represents m orthogonal basis vectors pj, j = 1, …, m and Λ diagonal matrics composed of eigenvalues λj, j = 1, …, m. To each eigenvalue λj belongs particular basis vector pj. Matrix of eigenvectors P is organized so that column vectors are sorted by decreasing magnitude of eigenvalues λ1<λ2<⋯<λm. In other words, eigenvectors are sorted by decreasing amount of information they provide.

Since HIS data contains many correlated features (i.e., spectral bands), the data set can be fully described by using only a subset of eigenvectors of covariance matrix Σ. The general assumption is that part of the information can be explained with k eigenvectors, which we call principal components. Various criteria can be used to determine the number of principal components, e.g., percentage of explained variance in the data. Mathematically we can write:(11)$$\frac{\sum_{j=1}^{s}\lambda_{j}}{\sum_{j=1}^{m}\lambda_{j}}\geq \epsilon$$
where ϵ represents predefined threshold. Usually it is set to 0.95 to keep 95% of initial variance of the data. Another option is to directly choose desired number of principal component. The covariance matrix can then be approximated by neglecting eigenvectors with small corresponding eigenvalues. In other words, we neglect $p_{j},$ where j=s+1,…,m. Approximated covariance matrix can then be calculated as:(12)$$\Sigma_{s}=\overset{s}{\underset{j=1}{\sum }}\lambda_{j}p_{j}p_{j}^{T}{}^{\;}$$
where vectors $p_{j}$, j=1,…,s define principal directions in which the data extends and is weighted by corresponding eigenvalues.

##### Linear Discriminant Analysis

Linear discriminant analysis is a robust classification method, but can also be used for dimension reduction and data visualization. Unlike PCA, which tries to maximize variance, it is a supervised machine learning method that computes decision boundaries which enhance the separation between multiple classes used. 

It tries to separate different classes by maximizing distances between projected means and minimizing projected variance. Both optimization problems are incorporated in one single criterion function which can be, for binary classification, written as:(13)$$\underset{w}{\max}\;\;J\left(w\right)=\frac{{\left(m_{1}-m_{2}\right)}^{2}}{s_{1}^{2}+s_{2}^{2}}$$
where $\;{\left(m_{1}-m_{2}\right)}^{2}$ represents the difference in means between the two classes and $s_{1}^{2}+s_{2}^{2}$ the total scatter (standard deviations) of the two classes. The goal of LDA is to find the vector w that maximizes criterion function $J\left(w\right)$. 

In LDA it is assumed that all *K* classes have equal covariance. Following this assumption we can obtain the following discriminant function for k-th class:(14)$$\delta_{k}\left(x\right)=x^{T}\Sigma^{-1}\mu_{k}-\frac{1}{2}\mu_{k}^{T}\Sigma^{-1}\mu_{k}+\log\pi_{k}$$

Which predicts the class with the highest value of δkx given an input x∈ℝp×1. In Equation (14) Σ∈ℝp×p represents common covariance matrix, μk∈ℝp×1 the mean of inputs for class k and πk prior distribution of class k. Symbol *p* represents number of attributes in each data instance.

Features are transformed so that classes are as separate as possible from each other and that features within a class are as close as possible. Transformed dimensions are ranked based on the separation ability. Maximal number of components must be at least one fewer than the number of classes used for classification. Therefore, since we performed binary classification in this study, only the first and only linear discriminant was used [38].

##### Partial Least Squares

Partial least squares is a technique that transforms the initial dataset to a reduced set of uncorrelated features using a technique similar to principal component analysis. It extracts features that describe maximum correlation with target variables; i.e., they provide the greatest predictive ability. This method is especially useful when features in initial dataset are highly collinear [39].

The underlying core equations of PLS could be written as:(15)X=TPT+E
(16)Y=UQT+F
where X∈ℝn×m is the matrix of independent variables (with hyperspectral data spectral bands) and Y∈ℝn×p is the matrix of dependant variables (these can be measured variables or dummy coded nominal variables). Symbol n represents number of data instances, m number of attributes in each independent variable and p number of attributes in each dependent variable. Matrices T, U∈ℝn×l respectively represent projections (scores) of X and Y. Matrices P∈ℝm×l and Q∈ℝp×l respectively represent orthogonal loadings matrices of X and Y. Symbol *l* represents a user-defined number of latent factors used in for regression. Model is optimized in such a way that the first score in X has maximum covariance with the first score in Y. Therefore, we can predict the first score in Y from the first score in X.

Partial least squares has some advantages over basic ordinary least square (OLS) solution. It is able to dispose correlated variables and model their shared and underlying information. In contrast to many machine learning methods, it can directly model multiple dependent variables at the same time. Several variants of PLS exist; we used Partial least squares discriminant analysis, which is an extension of PLS regression, the foundation for other variants.

##### ReliefF

ReliefF is an extension of the basic Relief algorithm, and is a generally well-performing attribute selector. It can provide a combined view of relevance and conditional dependencies between attributes. The algorithm prescribes a separate weight ($w_{j}^{\;}$) to each attribute, where higher values correspond to more important attributes. The basic idea of the algorithm is that is penalizes attributes which provide different result of the same class in comparison with its nearest neighbours [40]. 

At the beginning, ReliefF sets all attribute weights w_j_^i^ to zero, these are then iteratively adapted. Then, it selects a random observation $x_{r}\;$and k-nearest observations for each class. All the weights are updated for each nearest neighbours $x_{q}$ by equations:(17)$$w_{j}^{i}=w_{j}^{i-1}-\frac{\Delta_{j}\left(x_{r},x_{q}\right)}{m}\cdot d_{rq}$$
(18)$$w_{j}^{i}=w_{j}^{i-1}+\frac{p_{y}{}_{q}}{1-p_{y}{}_{r}}\cdot \frac{\Delta_{j}\left(x_{r},x_{q}\right)}{m}\cdot d_{rq}$$
where $w_{j}^{i}$ represents the weight of the j-th attribute at iteration i, m is the total number of iterations, $p_{y}{}_{q}\;$and $p_{y}{}_{r}$ are prior probabilities of classes where $x_{q}$ and $x_{r}\;$ respectively belong, and $d_{rq}$ is the distance function, which is subject to scaling. Symbol $\Delta_{j}\left(x_{r},x_{q}\right)$ represents the difference between prediction values of observations $x_{r}\;$ and $x_{q}$ for j-th attribute $F_{j}$. For continuous attributes it is calculated as:(19)$$\Delta_{j}\left(x_{r},x_{q}\right)=\frac{\left|x_{rj}-x_{qj}\right|}{\underset{\;}{\max}\left(F_{j}\right)-\underset{\;}{\min}(F_{j})}$$

##### RandomForest

Random forest is an ensemble technique that combines multiple de-correlated decision trees. Decision trees are fitted on various randomly chosen subsets of a given dataset. Overall performance of the model is increased by aggregating predictions from all trees and performing a majority vote for each class in classification problems. 

In the training phase of Random forest a technique called bootstrap aggregation or bagging is used.

Given training data instances in a matrix X∈ℝn×m and corresponding labels y∈ℝn×1 (where *n* represents number of data instances and *m* number of attributes in each instance), bagging repeatedly selects random data instances with replacement and fits B decision trees f  to these instances. Unseen data instances x′ are predicted by averaging all predictions made by individual decision trees:(20)y^=1B∑b=1Bfbx′
where y^ represents approximated predicted output. Bagging decreases variance without increase of bias. This leads to more accurate performance even if each individual decision tree is highly sensitive to noise. Furthermore, Random forest also includes feature bagging, i.e. selection of a random subset of the attributes in the training set. A small number of attributes may have a very strong prediction power for the response variable. Consequently, these attributes would be selected many times causing decision trees to become correlated. We used the Gini index as split criterion and for assessing variable importance. For each Random forest 100 trees were constructed [41].

##### Extreme Gradient Boosting

Gradient boosting is one of the most powerful and flexible machine learning methods, which can be applied to various machine learning problems. It refers to a class of ensemble methods used for predictive modelling problems. Similarly to Random forests, it is constructed from decision tree models (weak learners). Unlike Random forest, weak learners are added one at a time to correct errors produced by prior decision trees. This type of error correction is called boosting, where models are iteratively trained with gradient descent optimization of any differentiable loss function. For instance, a squared error may be used for regression problems and logarithmic loss for classification problems. New decision trees are trained on error residuals produced by initial learner. Intuitively, newly trained models are influenced more by misclassified observations or by areas where they are performing poorly. The contribution from all decision trees are aggregated to make the final prediction [42]. 

Simplified optimization could be mathematically written as follows. First model is initialized with a constant value with minimization of loss function Lxi, yi,θ:(21)f^0X=argminθ∑i=1nLxi, yi,θ
where X∈ℝn×m is a matrix of input data instances with corresponding labels y∈ℝn×1. Symbols n and m represent number of data instances and number of attributes in each instance, respectively. Based on the weak learner from the previous iteration, gradients and hessians are calculated and then a new weak learner is fitted using optimization problems. At the end of iteration, the model transfer function is updated as:(22)f^mX=f^m−1X+f^mX
where m=1,2,…,M, where *M* is the total number of weak learners. Unseen data instances x′ are then predicted by summation of all predictions made by individual decision trees:(23)y^=f^ x′=f^Mx′=∑m=0Mf^mx′

### 2.4. Support Vector Machines

Extracted or selected features from dimensionality reduction were pipelined to support vector machine classificator. Classification models were therefore built on reduced data sets, consisting of only the most prominent feature. In this study performance is tested on radial-basis kernel function for data transformation [43]. Hyperparameter tuning (gamma and C) was performed using a grid search, whereupon combinations yielding the best accuracy were retained.

Trained SVM classifier was evaluated using mean accuracy. It was iteratively trained and tested 5 times for each DR algorithm. Accuracies from all iterations were then averaged. For this reason an objective criteria is devised for comparison between all DR methods:(24)$$\begin{array}{c} {\bar{c}}^{\left(m\right)}=\frac{1}{F}\frac{1}{P}\overset{F}{\underset{f=1}{\sum }}\overset{P}{\underset{p=1}{\sum }}\Gamma \left(y_{p},{\hat{y}}_{p}^{\left(m\right)}\right)\\ \Gamma \left(y_{p},{\hat{y}}_{p}\right)=\left\{\begin{array}{cc} 1; & \mathrm{if}\;y_{p}={\hat{y}}_{p}^{\left(m\right)}\\ 0; & \mathrm{else} \end{array}\right.\\ y_{p},{\hat{y}}_{p}^{\left(m\right)}\in \left\{\mathrm{Healthy},\mathrm{Inoculated}\right\} \end{array}$$

In Equation (24) mean accuracy is labelled as ${\bar{c}}^{\left(m\right)}$. Superscript represents m-th DR method belonging to a set $M_{m}\in \left\{\mathrm{PCA},\;\mathrm{LDA},\;\mathrm{PLS},\;\mathrm{RF},\;\mathrm{RFF},\;\mathrm{XGB}\right\}$. Mean accuracy is calculated from comparison between predicted ${\hat{y}}_{p}$ and known $y_{p}$ labels of potato tuber, which can be either healthy or inoculated. It is calculated on test feature vectors for P predictions and F folds. In our case F=5 and P=12. Precision, recall and F1-score were calculated using equations in [44]. All analyses were performed in Python, using libraries scikit-learn [45] and XGBoost [46]. 

## 3. Results and Discussion

The first extracted features, or limited set of selected features, proved to be sufficient for accurate detection of infested potato tubers. Spectral differences between inoculated and healthy tubers were more pronounced in images of their outside (Figure 4). Spectral signatures of the outside of tubers show a larger variability in infested tubers, than in healthy ones. These differences are more pronounces in the SWIR region, where inoculated tubers uniformly exhibited higher reflectance than healthy tubers. In contrast, data from inside tubers shows comparatively little variability, regardless of inoculation status. The high variability in outside images could be a consequence of tuber surface characteristics. Healthy tubers are comparatively smooth, while galls cover the surface of inoculated tubers. This leads to a more varied viewing geometry, which was accounted for in pre-processing of the images.

We used the first two principal components for data visualization in a generated feature space. Feature vectors were separated into 5 cross-validation folds, and PCA performed on each fold of the training data, and applied to both train and test sets. The generated features from all folds were pooled to generate scatter plots of the first two PCA components (Figure 5). The first two components explain more than 80% of the variance in the data (93% for outside, and 84% for inside tubers). These scatter plots show a better distinction between healthy and inoculated tubers for data from the outside of tubers. Yet any linear separability does not appear to be present, at least not in the first two PCA components. 

Females of root-knot nematodes reside within a few millimeters below tuber skin, in the vascular ring, where they form comparatively large egg-sacs [47]. When these grow enough, they form galls on tuber surface. Even though some evidence exists that RKNs change the chemistry of the entire tuber, i.e., also the starchy insides [2], we expected classification success to be higher in outside data. This hypothesis was confirmed for all DR methods (Figure 6). All methods achieved a mean accuracy of at least 0.6 (Table 1 and Table 2). LDA showed the most consistent results, as it achieved a mean accuracy of 1.0 in both inside and outside tuber data. Overall RF came second, with XGB yielding very similar accuracies. PLS came next, followed by PCA, and lastly ReliefF achieved the worst results. Data was also analysed without any DR. Even though this method was capable of achieving good accuracy, a mean of 0.9 from both data sets, it is computationally much more burdensome. With SVMs solving the quadratic problem involves inverting the kernel matrix, with a complexity of up to n^3^, where n is the feature space [43]. RandomForest, XGB and ReliefF suffer from the same problem, as they only select features, i.e. they generate a subset of the original feature space. Furthermore, even though all three feature selection methods are robust, they can suffer from overfitting and should be optimized accordingly [48]. In this regard feature extraction methods are beneficial, since they generate a new feature space, with lower dimensionality. 

Increasing the number of features has an expected effect, of increasing classification accuracies (Figure 7). The most profound effect is observable in PCA and RFF, while RF shows the smallest change. Unlike the other five methods, only one feature gets extracted by LDA in binary classification. Interestingly, PCA on inside data decreases accuracy with the first three components. Accuracy then increases with more features, but still remains bellow PLS and LDA accuracies. With more features extracted, only PLS and XGB achieve a 100% accuracy in both inside and outside tuber data. Our results indicate that even with an extreme reduction, to just one feature, identification accuracies are still acceptable to excellent (mean accuracy between 0.8 and 1.0).

Compared to no DR, dimensionality reduction using PLS and PCA on outside data reduced detection accuracy (Table 2). On the other hand, in a data set with less pronounced patterns, such as tuber insides, classification accuracy was increased by using PLS. One of the purposes of dimensionality reduction is to generate or retain only those features, which are informative for the problem under study. This way the signal-to-noise ratio can be improved, leading to better model performance.

Dimensionality reduction algorithms use different metrics to asses feature importance. For example, principal components analysis uses a correlation matrix between generated components and original features. Correlations above or below 0.7 or −0.7, respectively, are considered as relevant. In PLS correlations can also be considered, but a more accurate assessment of feature importance is possible using variable importance in projection analysis (VIP) [49]. VIP coefficients reflect the relative importance of each variable for each variate in the prediction model. Variable importance in LDA was calculated as LDA scalings, i.e., the eigenvectors of the components. Important to note here is that LDA is a discriminant analysis method and as such it maximizes the between-group variance. The eigenvectors of the comparison matrix of between and within group’s sum of squares and cross-products describe how much the original variables contribute to the new component(s). Gini importance, used with RandomForests and Extreme gradient boosting, provides a relative ranking of the original features, and is a by-product of the training of the classifier [50]. Lastly, ReliefF assigns feature relevance depending on the difference between this feature and two neighbours of the same and opposite classes [51]. Each of these methods provides their own metric of variable importance. In order to directly compare all methods, we normalized their values to a range of 0 to 1.

RandomForest and XGB identified several relevant wavelengths, distributed comparatively evenly throughout the spectrum, in both data sets. Similarly, LDA also identified a large number of relevant wavelengths, but unlike RF, these were not evenly distributed in the outside data set. In this set the importance of variable shifted towards the SWIR region. Only ReliefF showed a different pattern, compared to the other five methods. Interestingly, while PLS, LDA, RF, and XGB show a similar grouping of relevant wavelengths in the same SWIR regions (1500–1600 nm, 1850–2000 nm, and 2300–2450 nm), PCA found relevant regions between these groups (1600–1800 nm, and 2100–2200 nm). On the other hand, ReliefF found relevant wavelengths in the range 1000–1400 nm, i.e., in shorter wavelengths than the other methods. With inside data, only PCA found relevant regions in the SWIR part of the spectrum, above 1500 nm (1600–1850 nm), while the remaining methods found a strong grouping of relevant wavelengths closer to the VNIR region, between 1000 and 1200 nm (Figure 8). This spectral region is linked to various hydrocarbons, both aliphatic and aromatic [52]. The region identified by PCA is also linked to different hydrocarbons (aliphatic, aromatic, and methyl), but also alcohols, amines and proteins. In the outside data set, regions linked to water, polysaccharides, aromatic amines (1850–2000 nm), and lipids and glucose (2300–2450 nm) were identified as relevant. In order to fully test the accuracy of the different variable importance measures, employed by the dimensionality reduction methods, more detailed chemometric analyses of potato tubers are needed. 

Differences on the inside of tubers are most likely of a chemical nature, while the outside is influenced by both differences in chemistry as well as surface texture. While surface texture effects can be reduced using normalization and Savitzky–Golay derivatives, they can still affect the classifications. The infested tubers used in this study had visible symptoms (bulges on the outside, approximately 3–5 mm in diameter), covering at most 50% of the surface. In our case, they added another distinguishing dimension, leading to better classification accuracy with data from tuber surfaces. Nevertheless, the feasibility of hyperspectral imaging for detection of latent (i.e., without visible symptoms) infestations with nematodes in potatoes has been shown by Žibrat et al. [2]. In that case surface texture didn’t influence spectral signatures; therefore the observed differences are exclusively due to differences in chemistry between healthy and infested tubers. From a practical application standpoint, even if tubers in production spot checks would have to be halved and imaged, the throughput of such a method would be much higher than with molecular analyses (e.g., real–time PCR). So the benefit of using hyperspectral imaging for detecting infestations, even with processed tubers, is evident.

Molecular spectra are the result of motions of atomic nuclei. They can rotate, vibrate, wag and move together or apart along a straight line (this type of movement is called stretching). Vibrations follow a functional description, i.e. the type of vibration determines the frequency at which it absorbs energy. The amplitude of absorption is determined by absorptivity and the number of molecules in the beam path of a particular sensor. Changes in spectral responses follow Beer’s Law, which states that the absorbance is equal to product of absorptivity of a molecule and the concentration of molecules. The above means that the light absorbed by plant tissue depends on the chemical composition of that tissue, the concentration of individual molecule species, and their interactions [52]. We’ve identified several groups of molecules, which account for the differences between healthy and infested potato tubers (such as aliphatic and aromatic hydrocarbons). The two spectral regions, VNIR and SWIR, are generally linked to different characteristics of plants, but there is some overlap. In the VNIR region we mostly get information about pigments and structure, e.g., morphological structure of plant leaves. In addition, in wavelengths above 700 nm there is also information related to hydrocarbons (mostly aliphatic) and alcohols, these are generally the third and fourth overtones of the C-H stretch. The SWIR region carries information on plant biophysical properties (e.g., hydrocarbons and proteins). Changes in chemistry can therefore be measured in both. Since we do not know how exactly nematode infestations change the chemistry of potato tubers, we decided to use both systems. The wider spectral range of the combined system enables us to better search for the effects of nematode infestations. With the currently available information we can only speculate which exact compounds account for the observed differences between inoculated and healthy potato tubers.

We identified several relevant wavelengths from each DR method. The latter use different methods for deciding, which wavelength is relevant, so we would recommend to look for overlaps between the methods. Those wavelengths which were identified as relevant by different methods are good candidates for a multispectral sensor. Different bandpass filters are commercially available, with different spectral ranges and bandwidths. So it might not be necessarily needed to develop new filters. Using these filters we would effectively get a multispectral data set. New classification models would then have to be developed using this reduced data. Using this procedure we could assess the importance of each new wide band and determine which ones significantly increase classification success. Potentially this would mean that by reducing a hyperspectral dataset to a few-band (~5 bands) multispectral one we would still get acceptable classification accuracy. Production of such a dedicated multispectral system would be much cheaper, than a hyperspectral one, with similar classification success. 

## 4. Conclusions

In this study we have shown that discrimination between healthy and inoculated potatoes with quarantine pests is possible based on hyperspectral image analysis. We successfully reduced the initial hyper-dimensional feature space to one–dimensional (or few–dimensional) with the use of dimensionality reduction algorithms, and still obtained high classification accuracies. These results suggest that a comparatively low–cost imaging system utilizing band–pass filters could be designed for the specific purpose of identifying tubers infested by root–knot nematodes. But the study was performed on a very small data set and further analyses are needed to fully test this idea.

## Figures and Tables

**Figure 1 sensors-22-00367-f001:**
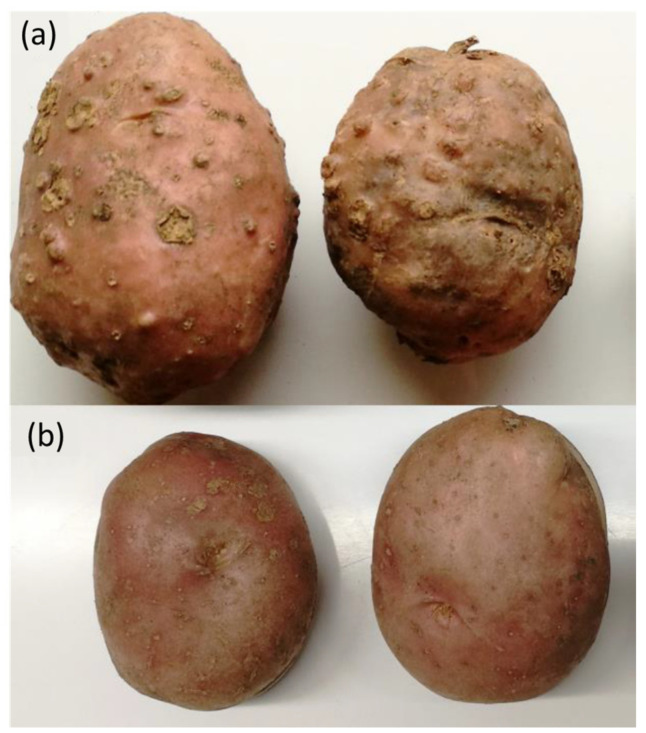
Inoculated and healthy potato tuber. (**a**) inoculated tubers, and (**b**) healthy tubers. Note the galls on the surface of the infested tubers. The difference in size is not necessarily symptomatic.

**Figure 2 sensors-22-00367-f002:**
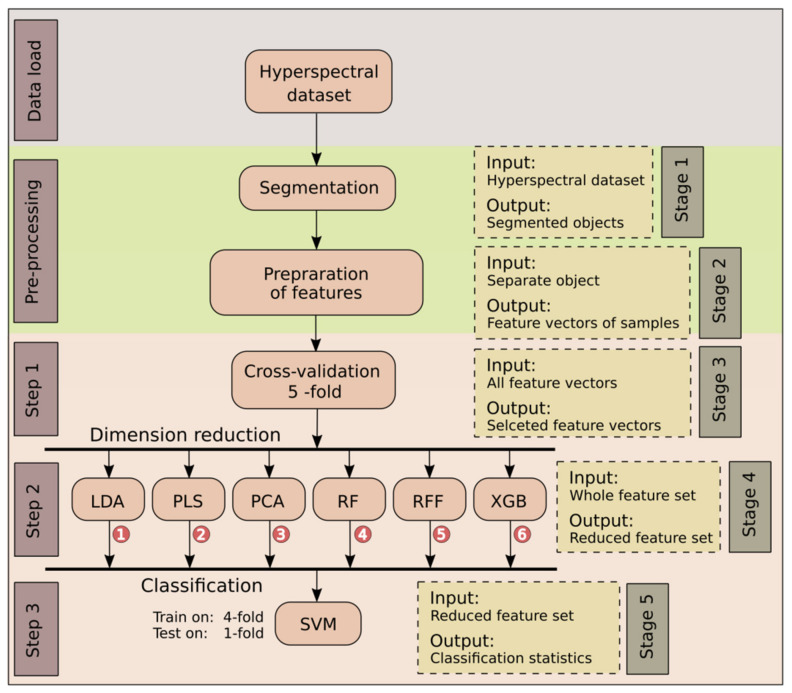
Analysis flowchart for proposed procedure. Data load: Image acquisition. Pre-processing: Image segmentation and feature preparation techniques. Step 1: Separation of training and validation dataset with cross-validation. Step 2: Dimensionality reduction with LDA, PCA, RF, RFF, XGB or PLS. Step 3: Classification with the use of SVM and 5-fold cross-validation.

**Figure 3 sensors-22-00367-f003:**
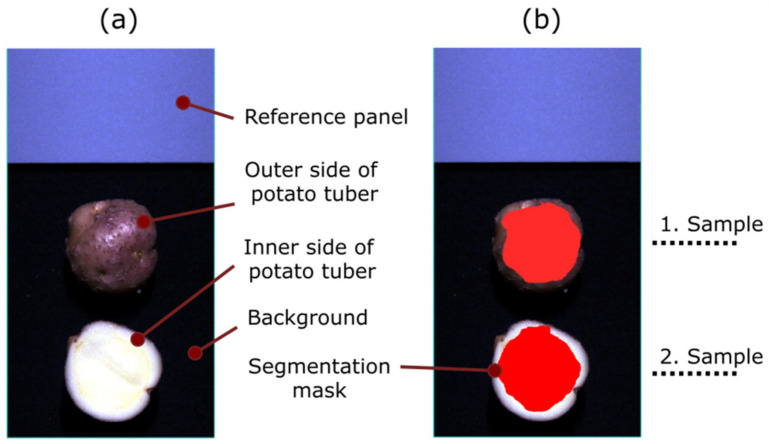
Hyperspectral image of tubers prior to segmentation. (**a**) With labelled segmentation class which construct the image. (**b**) Same image with applied segmentation masks of potato tubers and labelled consecutive sample number.

**Figure 4 sensors-22-00367-f004:**
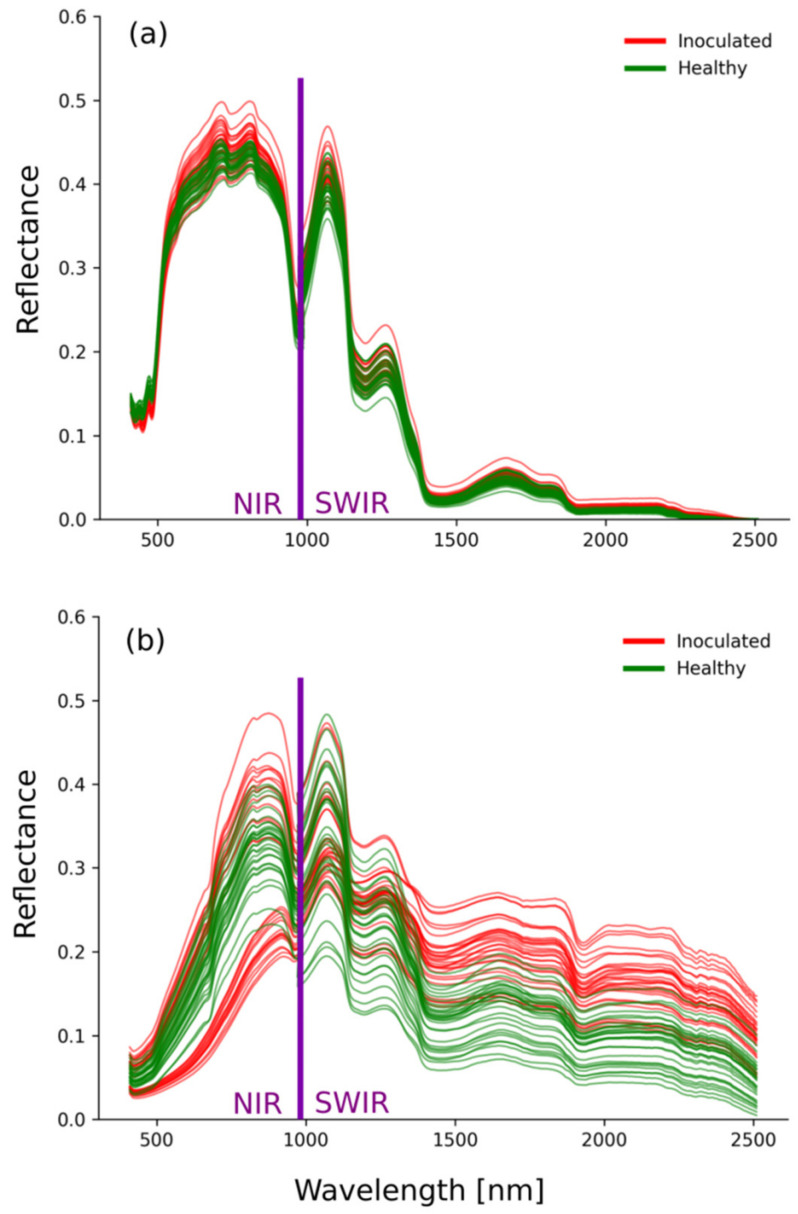
Reflectance feature vectors for (**a**) inside and (**b**) outside of potato tuber. Green colour refers to healthy and red to inoculated specimens. Separation between VNIR and SWIR cameras is marked with violet colour.

**Figure 5 sensors-22-00367-f005:**
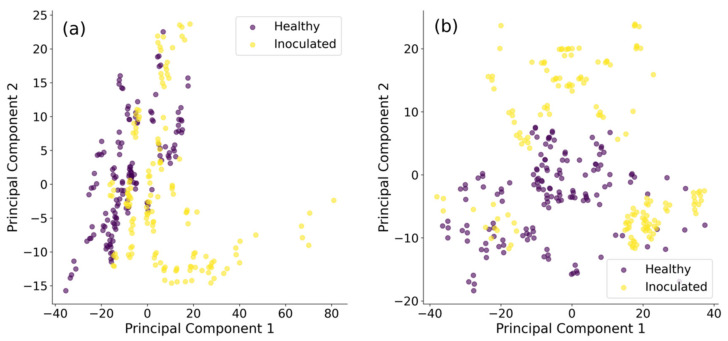
Scatter plot of first two principal components for (**a**) inside and (**b**) outside of potato tubers. Yellow colour refers to healthy and purple colour to inoculated specimens.

**Figure 6 sensors-22-00367-f006:**
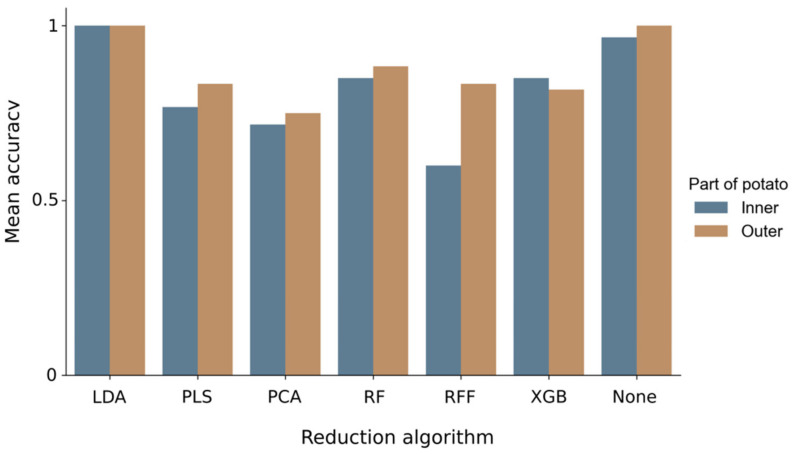
Mean accuracy for several dimensional reduction algorithms. Accuracy of classifier trained on data from: outer side of potato (orange columns), inner side of potato (blue columns).

**Figure 7 sensors-22-00367-f007:**
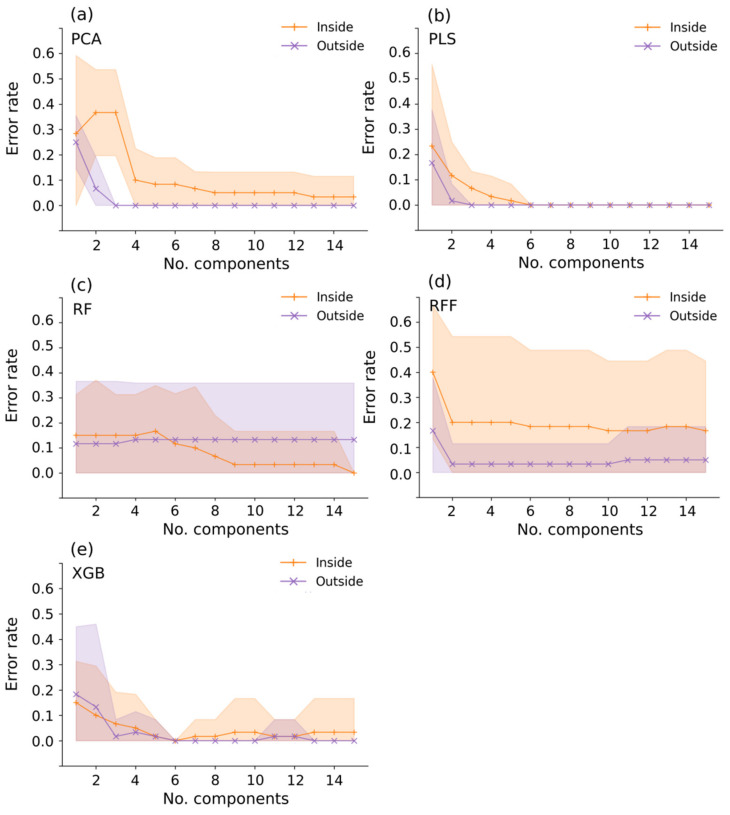
Influence of increasing number of features on classification error rates for both inside and outside tuber data. Since LDA generates only one feature in binary classification it wasn’t included in this figure. (**a**) Principal component analysis, (**b**) Partial least squares, (**c**) Random forest, (**d**) ReliefF, and (**e**) Extreme gradient boosting.

**Figure 8 sensors-22-00367-f008:**
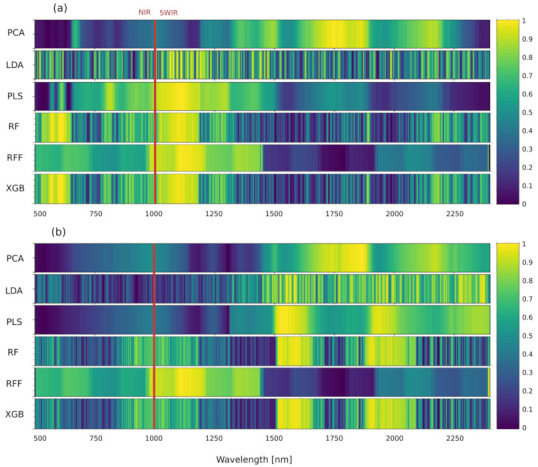
Importance of individual wavelengths, as determined by the different DR methods. (**a**) inside, and (**b**) outside of potato tuber. Brighter colours represent higher importance. Separation between VNIR and SWIR cameras is marked with red color.

**Table 1 sensors-22-00367-t001:** Comparison of classification results with only one feature for selected DR methods for outside tuber data.

Outer Side of Potato Tuber					
Method	Class	Precision	Recall	F1-Score	Accuracy
LDA	Healthy	1.00	1.00	1.00	1.00
	Inoculated	1.00	1.00	1.00	
PLS	Healthy	0.88	0.77	0.82	0.83
	Inoculated	0.79	0.90	0.84	
PCA	Healthy	0.86	0.60	0.71	0.75
	Inoculated	0.69	0.90	0.78	
RF	Healthy	0.93	0.83	0.88	0.88
	Inoculated	0.85	0.93	0.89	
RFF	Healthy	0.88	0.77	0.82	0.83
	Inoculated	0.79	0.90	0.84	
XGB	Healthy	0.81	0.83	0.82	0.82
	Inoculated	0.83	0.80	0.81	
None	Healthy	1.00	1.00	1.00	1.00
	Inoculated	1.00	1.00	1.00	

**Table 2 sensors-22-00367-t002:** Comparison of classification results with only one feature for selected DR methods for inside tuber data.

Outer Side of Potato Tuber					
Method	Class	Precision	Recall	F1-Score	Accuracy
LDA	Healthy	1.00	1.00	1.00	1.00
	Inoculated	1.00	1.00	1.00	
PLS	Healthy	0.75	0.80	0.77	0.77
	Inoculated	0.79	0.73	0.76	
PCA	Healthy	0.70	0.77	0.73	0.72
	Inoculated	0.74	0.67	0.70	
RF	Healthy	0.89	0.80	0.84	0.85
	Inoculated	0.82	0.90	0.86	
RFF	Healthy	0.64	0.47	0.54	0.60
	Inoculated	0.58	0.73	0.65	
XGB	Healthy	0.89	0.80	0.84	0.85
	Inoculated	0.82	0.90	0.86	
None	Healthy	0.94	1.00	0.97	0.97
	Inoculated	1.00	0.93	0.97	

## Data Availability

The data presented in this study are available on request from the corresponding author. The data are not publicly available due to their use in ongoing research projects.

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
