# Peer review of "Comparison of Selected Dimensionality Reduction Methods for Detection of Root-Knot Nematode Infestations in Potato Tubers Using Hyperspectral Imaging"

_sensors, 2022, doi:10.3390/s22010367_

Round 1
Reviewer 1 Report
This article provides an interesting read on the various DR methods used to identify RKNs in potatoes. Both the agriculture and food industry would definitely benefit from a low-cost technique that could identify these pests with high accuracy. My main concern surrounding this article is the small data set however the authors have addressed this and I would suggest following this publication up with another supporting these results with a much larger dataset (if possible). I am happy to recommend this article for publication providing these specific things are addressed:
1: I would not use a reference number to replace text. For example, on line 35 it states 'described by [3]'. Please replace this with 'described by Carneiro et. al.' and then have the reference in the appropriate place (usually at the end of the sentence). This can also be seen on lines 86, 89, 91, 95, 123 and 241.
2: On line 116 please clarify if it is 20 potato plants that are one-day old (I'm assuming that it is), or potato plants that are 20 days old. The phrase 20-day-old plants can be confusing.
3: Section 2.1 Hyperspectral image acquisition should be 2.2.
4: Line 156 the authors state 'five steps' as shown in Figure 2. Figure 2 then explains the process as data load, pre-processing then steps 1,2 and 3. I would either change Figure 2 so it has specifically 5 steps of change the wording in line 156.
5: Check for consistency as section header 2.3.1 is bold and 2.3.2 is not.
6: In the materials and methods where the equations are included please add next to each equation the equation number. In the text on lines 206, 233, 235 and 301 the authors refer to specific equation numbers but they are not numbered so can be unclear.
7: Line 208, try-and-error should be trial-and-error.
8: At the first instance please define what NIR and SWIR stands for.
9: Figure 4, for consistency I would capitalise your legends of inoculated and healthy as these words are capitalised throughout the rest of the article.
10: The materials and methods section has quite a lot of detail on the segmentation but lacks greater detail on the different DR methods used. I suggest expanding on the sections that describe PCA, LDA, PLS and RF.
Reviewer 2 Report
This paper concentrate on the use of hyperspectral imaging (HSI) combined with the adoption of dimensionality reduction for detecting the occurrence of root-knot nematode infestation in potato tuber. Three feature extraction (LDA, PLS, PCA) and one feature selection (Random Forest) are compared and evaluated for classification of inoculated (diseased) and healthy tubers. The classification model was built using SVM classifier. From the experiments, the author concluded that the LDA dimensionality reduction method performs the best for classification for both the inside and outer sides of potato tuber hyperspectral images. The paper is generally easy to read. However, several issues need to be fixed before I can recommend it for publication.
Please describe the mechanism of why hyperspectral data is relevant for detecting plant disease on the tuber. The mechanism is more or less of what you see when people applied HSI to detect disease on plants (based on leaf reflectance – like the reference 1 and 10). In the tuber case, please specify what likely has changed, it is due to physical properties, texture attributed to the tuber, or even due to chemical constituents. Please relate this change mechanism with the spectral changes. Apart from reviewing the HSI technique to detect plant disease from leaf reflectance, give also additional review that applies HSI to detect diseases on root or tuber or tissue.
The experiment test on a very limited sample, only 10 samples, 8 samples are assigned as training, remaining 2 as test samples. I suggest that you can repeat this experiment using different approaches like using a classification based on a super-pixel scale. For each sample, you can group the spectra based on super-pixel. Therefore, you will have many samples to test the effectiveness of the DR method on your classification task.
Line 83-91: you state the current trend shows that many researchers tend to use feature selection over feature extraction. In your work, you concentrate much on feature extraction. Right now, you are comparing three feature extraction and one feature selection. I suggest you add at least one more feature selection technique (the recent technique) so that the comparison is a bit strong (fairer).
Line 103 – 105: You state your first hypothesis: DR methods can achieve good classification accuracy even with only one feature. – This hypothesis needs to be verified by showing the impact of reducing the feature up to the minimal features (one feature). The current manuscript did not provide this information. It is good to provide the analysis based on the different number of selected features. Second hypothesis: Data from the outside of tubers will achieve better classification success. Why are you interested to compare the HS image of the outside and inside tuber? The most interesting reason why people apply HSI is due to its capability to deliver non-invasive and non-destructive plant disease detection (you state some in your abstract). The main advantage is that people just need to scan the target material (tuber) without destroying the tissue. Otherwise, you need to provide information or explanation of why examining the inner tissue is important for plant (potato) protection.
Line 141-145: In your method, you use 2 sets of HSI cameras the VNIR and SWIR. What is the number of bands of each camera? What is the individual wavelength range for VNIR-1600 camera and SWIR-384 camera? Did you use all of the wavelengths, or did you discard some noisy bands? – Noisy band is usually present at the beginning and at the far end of the spectra reflectance.
In your work, what is your motivation to combine the spectra from VNIR and SWIR? Which (biochemical and biophysical) properties you are interested to see the modification due to stress? Since Different wavelength region is only useful for certain traits (properties), please guide the reader on why you think combining both VNIR and SWIR will be beneficial.
Another concern is that you state earlier in (line 60-61) that Hyperspectral data shows a high level of collinearity between spectral bands, leading to high redundancy and decreasing the signal-to-noise ratio. Therefore, before combining the VNIR and SWIR, it will be interesting to see the effect (any improvement in classification accuracy?) when you perform the dimensionality reduction on each individual VNIR and SWIR spectral band? As you show the spectra profile in Figure 4, each VNIR and SWIR has its unique pattern for discrimination purposes. It would be interesting to perform DR on each spectral region separately as this will make it easy to relate the important features to the spectral properties of the tubers. After doing this approach, if you don’t see any improvement happen, then it is a good option by performing the DR based on the combined VNIR+SWIR spectra. If you can see good improvement by applying DR on VNIR spectra, this will be great because the cost of using a VNIR camera is much cheaper compares to SWIR.
Line 343: You state that “This hypothesis was confirmed for all DR methods (Figure 5)”. But, Figure 5 was provided to explain the PCA, not PLSR LDA and RF?
Line 390-395: You observe from the DR method some wavelength regions are more prominent. Can you provide the relationship in terms of physical or biochemical properties that modify the spectral band?
Line 415-416: You stated that the experimental results suggest that a comparatively low-cost imaging system utilizing bandpass filters could be designed for the specific purpose of identifying tubers infected by root-knot nematodes. Could you describe a bit more the idea of this implementation? Did you mean to design an additional filter to the spectrograph of the HS camera? Since using different DR methods has uncovered different key spectral bands, which are the most appropriate band to select for the purpose of the bandpass filter?
Round 2
Reviewer 1 Report
I am happy with the changes that have been made. No further comments from myself.
Reviewer 2 Report
All of my previous concerns have been addressed properly.I do not have further concerns.